# Participatory Mapping of Demand for Ecosystem Services in Agricultural Landscapes

**Carmen Schwartz [1,2,\*], Mostafa Shaaban [1], Sonoko Dorothea Bellingrath-Kimura [1,2] and Annette Piorr [1]**

1   Leibniz Centre for Agricultural Landscape Research (ZALF e.V.), Eberswalder Straße 84,
    15374 Müncheberg, Germany; mostafa.shaaban@zalf.de (M.S.); belks@zalf.de (S.D.B.-K.); apiorr@zalf.de (A.P.)
2   Thaer-Institute of Agricultural and Horticultural Sciences, Humboldt-University Berlin, Unter den Linden 6,
    10117 Berlin, Germany
\*   Correspondence: carmen.schwartz@zalf.de

**Abstract:** Agricultural land use systems have been optimized for producing provisioning ecosystem services (ES) in the past few decades, often at the expense of regulating and cultural services. Research has focused mainly on the supply side of ES and related trade-offs, but the demand side for regulatory services remains largely neglected. The objective of this paper is to evaluate the usefulness of participatory geographic information system (PGIS) methods for demand assessment in larger rural and agrarian contexts by identifying spatially explicit demand patterns for ES, thereby enlarging the body of participatory approaches to ES-based land use management. Accordingly, we map, assess, and statistically and spatially analyze different demands for five ES by different stakeholder groups in agricultural landscapes in three case studies. The results are presented in a stakeholder workshop and prerequisites for collaborative ES management are discussed. Our results show that poor correlation exists between stakeholder groups and demands for ES; however, arable land constitutes the highest share of the mapped area of demands for the five ES. These results have been validated by both the survey and the stakeholder workshop. Our study concludes that PGIS represents a useful tool to link demand assessments and landscape management systematically, especially for decision support systems.

**Keywords:** participatory mapping; ecosystem services; demand; PGIS; agricultural landscapes



## 1. Introduction

Agricultural systems are genuinely social–ecological systems, with the possibility of producing a wide variety of provisioning ecosystem services (ES) and providing key ecological processes and regulatory services. The magnitude of the supply of agricultural ES is influenced by the interactions between the social and ecological systems, i.e., the farmer, current political regulations, consumer choices and the farming ecosystem [1,2].

Agricultural areas have been characterized by intensification, mechanization and a reduction in the labor force in the past few decades [3]. While this process has been considered essential for achieving food security, regulating ES have been mainly negatively affected by this process, such as pollination, agrobiodiversity, water cycling and clean air [4]. The continuous process of concentration of large parts of the land in the hands of few owners in the North-east of Germany has led to an increase in the average field size with increasing attention paid to maximizing the production of provisioning services, often reached through the reduction of landscape elements, such as tree rows and hedges, with negative consequences for regulating ES [5].

The transition to a more sustainable form of land use must fully account for the economic, ecological and social implications of agricultural productivity. The services and dis-services generated by these systems affect the stability of local and global ecosystems and, by extension, the people living in these systems [2].

Scientifically displaying the value of ES in agricultural landscapes has gained increased attention in the past few years. Efforts have been undertaken to display the biophysical, economic, environmental and social value of land use systems in monetary [6,7], non-monetary [8] and spatially explicit ways [9]. The Common International Classification of ES (CICES) [10] is a widely used assessment framework. It lists three main categories of ES: (1) provisioning, (2) regulation and maintenance and (3) cultural (see examples for each category in Table 1). These services are generated by underlying structures, processes and functions of the ecosystems. Biodiversity is the diversity of all living organisms and is considered to be both a function service that many other processes and services depend on and a service because it has direct benefits to human well-being. Our study focuses on one provisioning service (biomass yield) and four regulatory services (biodiversity, carbon sequestration, erosion control and water availability).

**Table 1.** Examples of provisioning, regulation and maintenance, and cultural ES modified according to CICES [10].

| Section | Division | Group | Class | Class Type | Simple Descriptor |
|---|---|---|---|---|---|
| **Provisioning** | Biomass | Cultivated terrestrial plants for nutrition, materials or energy | Cultivated terrestrial plants (including fungi, algae) grown for nutritional purposes | Crops by amount, type (e.g., cereals, root crops, soft fruit) | Any crops and fruit grown by humans for food; food crops |
| | Biomass | Cultivated terrestrial plants for nutrition, materials or energy | Cultivated plants (including fungi, algae) grown as a source of energy | By amount, type, source | Plant materials used as a source of energy |
| **Regulation and Maintenance** | Regulation of physical, chemical and biological conditions | Regulation of baseline flows and extreme events | Hydrological cycle and water flow regulation (including flood control and coastal protection) | By depth/volumes | Regulating the flows of water in our environment |
| | Regulation of physical, chemical and biological conditions | Pest and disease control | Pest control (including invasive species) | By reduction in incidence, risk, area protected by type of living system | Controlling pests and invasive species |
| **Cultural** | Direct, in situ and outdoor interactions with living systems that depend on presence in the environmental setting | Physical and experiential interactions with natural environment | Characteristics of living systems that that enable activities promoting health, recuperation or enjoyment through active or immersive interactions | By type of living system or environmental setting | Using the environment for sport and recreation; using nature to help stay fit |
| | Direct, in situ and outdoor interactions with living systems that depend on presence in the environmental setting | Intellectual and representative interactions with the natural environment | Characteristics of living systems that are resonant in terms of culture or heritage | By type of living system or environmental setting | The things in nature that help people identify with the history or culture of where they live or come from |

## 1.1. Mapping & PGIS Approaches

Participatory mapping of ES has gained increased attention in urban and rural contexts in the last few years [11–13]. Participatory geographic information system (PGIS) tools have been used to involve stakeholders in spatially explicit ES assessments by combining survey questions with a mapping component. They have proven useful for engaging people and their knowledge of landscapes in identifying and valuing ES in direct relationship to the

landscape they originate from or are provided in. The PGIS tools have been used for highlighting the spatial heterogeneity of ES [14], perceived trade-offs and synergies [15], and flows of ES [16].

### 1.2. Demand and Supply Assessments and Trade-Offs

Identifying both an ecosystem's capacity to provide services (the supply side) and the social demand for those services (the demand side) remains a challenge in ES research [17]. Few studies combine assessments of both demand and supply of ES within the same study and region [18–20]; therefore, limited evidence exists on the demand for ES in relation to the supply within the same area. Spatially explicit knowledge about demand can show the connectivity between ecosystems and the beneficiaries of their services, and can predict competition over resources or possibilities for cooperation [21]. Combined approaches of supply and demand can capture the biophysical conditions for ES supply and societal needs that influence the actual supply that can be addressed via participatory mapping.

Geijzendorffer et al. [22] developed a scheme that encompasses five interlinked components along the supply demand continuum: interest, demand, match, managed supply and potential supply. An interest in ES becomes a demand only through the actual allocation of scarce resources, such as time or money, to fulfil this interest in a specific area and time. What follows from this definition is the identification of three types of mismatches between demand and supply: (i) actual uptake of ES is higher than the ecosystem can sustainably supply, (ii) managed supply leads to the production of certain services at the expense of others, trade-offs occur, and (iii) demand is unsatisfied due to insufficient supply. This scheme allows for the identification of trade-offs that a simple overlay of demand and supply maps might miss [23]. Some studies display trade-offs and synergies as a balance for the overall spatial entity under consideration [24], but studies that display trade-offs in a spatially explicit context are rare. Such information would be crucial for regionally optimized decision-making as different ES, due to biophysical and environmental processes and flows, are characterized by different spatial extensions. The trade-offs identified depend on the valuation method with which ES are assessed [25]. The dominance of biophysical and monetary supply assessments leads to a potential bias in the trade-offs identified. Therefore, including trade-offs in the demand assessments has the advantage of identifying trade-offs that would escape in a (biophysical or monetary) supply assessment. We visualize the preferences for ES in the demand assessment by different actors depending on the perceived utilities they expect from these ES. Trade-offs between demands are visualized when they are mapped in a spatial context, where the actors make decisions regarding allocating and specifying the areas of demands. Bringing the spatial ES supply demand trade-offs together helps one to identify the decision behaviors of the actors in response to the ecological behavior. It can help potential trade-offs to be foreseen that can arise if several ES are demanded but cannot be supplied in the same area. Furthermore, it includes a beneficiary perspective on trade-offs and can be a way of identifying potential bundles of ES that people demand in the same area with implications for management that reduces trade-offs between selected ES.

### 1.3. Assessment of Regulatory Services

A genuine problem with accounting for regulatory services is that they are nonmarketable goods and services and thus stay invisible if demand is assessed only for goods consumed. However, regulating ES are essential for the maintenance and perpetuation of the entire ecosystem. Regulating ES encompass a wide range of services closely linked to functions and processes that sustain the existence of the ecosystem itself and all living entities contained within it [26]. Regulating and cultural services are often traded off against the production of food, fiber and fuel [27]. Assessing regulating ES can be a preventive measure for avoiding trade-offs. Most demand assessments focus on provisioning services by assessing the marketed quantity (such as the yield of agricultural or forestry goods), while some on cultural services assess the willingness to pay or the distance travelled

to visit natural environments (i.e., recreation). Regulatory services are more difficult to assess, and so are their benefits on human well-being that might occur with spatial or temporal delay or might not be perceivable at all. The more complex relationship between regulatory services and human well-being is one reason for a striking demand assessment gap in this field [28]. Furthermore, beneficiaries might not be aware of their demand for regulatory services, which constrains the possibility of assessing it [29]. Defining clear beneficiary groups in advance helps to avoid spatial mismatches between ES and their beneficiaries [21].

With this background from the literature, we were able to identify four main research gaps: (1) a lack of translation of trade-offs into the spatial dimension. ES assessments have become precise in identifying gaps of supply of singular and multiple ES. However, trade-offs between ES, especially regulating ES, become visible only if displayed in a spatially explicit context; (2) a lack of sufficient differentiation by beneficiaries of ES. Demands are directly related to the people benefiting from the supply. In the case of regulating ES, beneficiaries can be the people living directly in the surrounding area of the ES produced (e.g., water regulation) or people benefitting from the global effects of climate regulation worldwide (e.g., carbon sequestration); (3) a lack of studies spatially assessing the demand for regulating and provisioning ES. Most studies using PGIS assess preferences for cultural ES. Empirical assessment of the demand for and understanding of regulating ES is missing; and (4) the lack of an empirical grounding of Geijzendorffer's concept regarding demand.

In order to address these gaps, our study combines the assessment of demand for selected regulation ES with a digital mapping exercise. The goal of this approach is to assess demands differentiated by stakeholder group, ES and region. We want to find out whether demands differ by stakeholder group, and if demands for different, non-synergetic ES show spatial overlay and thus cause conflicts in land use decisions. We investigate supply perceived and demand stated, thereby analyzing the gap between the state perceived and the state desired. We choose a stakeholder-based approach to relate the demand formulated directly to specific groups of beneficiaries. In our study, we furthermore aim to include the stated interest in relation to the perceived current supply in a spatially explicit location for all ES individually. Thereby, we capture the interest formulated by individuals and relate it to the potential supply perceived by the participants.

We propose an approach of combining participatory mapping with a questionnaire that allows the evaluation of the perceived current supply and demand for several ES. The supply of ES in agricultural landscapes depends on the configuration of the site, and increases with extent [30]. Therefore, we combine the evaluation of ES with a mapping component that allows the mapping of large areas, sub regions and single plot areas. Furthermore, we discuss the areas highlighted in the mapping exercise in a stakeholder workshop. With this approach, we aim at obtaining information about possible agricultural areas of interest for ES management on a landscape level.

Our research questions are: Do demands for ES differ between stakeholder groups? Is there a relationship between demand for ES and current land use? Can we identify "hot spots" of demand for ES?

## 2. Materials and Methods

### 2.1. Study Regions

We selected three case study areas (CSAs) located in three administrative NUTS3 districts of Brandenburg, a federal state in the northeast of Germany. The districts were chosen according to different degrees of heterogeneity in soil composition, natural vegetation and different types of land use, displayed and visualized in Table 2 and Figure 1.

**Table 2.** Land use in all case study areas, 2019 (MOL = Märkisch-Oderland; OPR = Ostprignitz-Ruppin; UM = Uckermark) [27].

|  | MOL | OPR | UM |
|---|---|---|---|
| Total Area (km$^2$) | 2158.5 | 2508.65 | 3058.35 |
| Agricultural Area (km$^2$) | 1255 (58.14% of total area) | 1253 (49.95% of total area) | 1766 (57.77% of total area) |
| Agricultural Area—Crops (km$^2$) | 1160 (92.43% of agricultural area) | 907 (72.39% of agricultural area) | 1472 (83.35% of agricultural area) |
| Agricultural Area—Grassland (km$^2$) | 91 (7.25% of agricultural area) | 341 (27.21% of agricultural area) | 293 (16.6% of agricultural area) |
| Perennial cultures and others (km$^2$) | 4 (0.32% of agricultural area) | 5 (0.32% of agricultural area) | 1 (0.05% of agricultural area) |
| Forest Area (km$^2$) | 510.27 (23.6% of total area) | 813.76 (32.44% of total area) | 748.11 (24.46% of total area) |

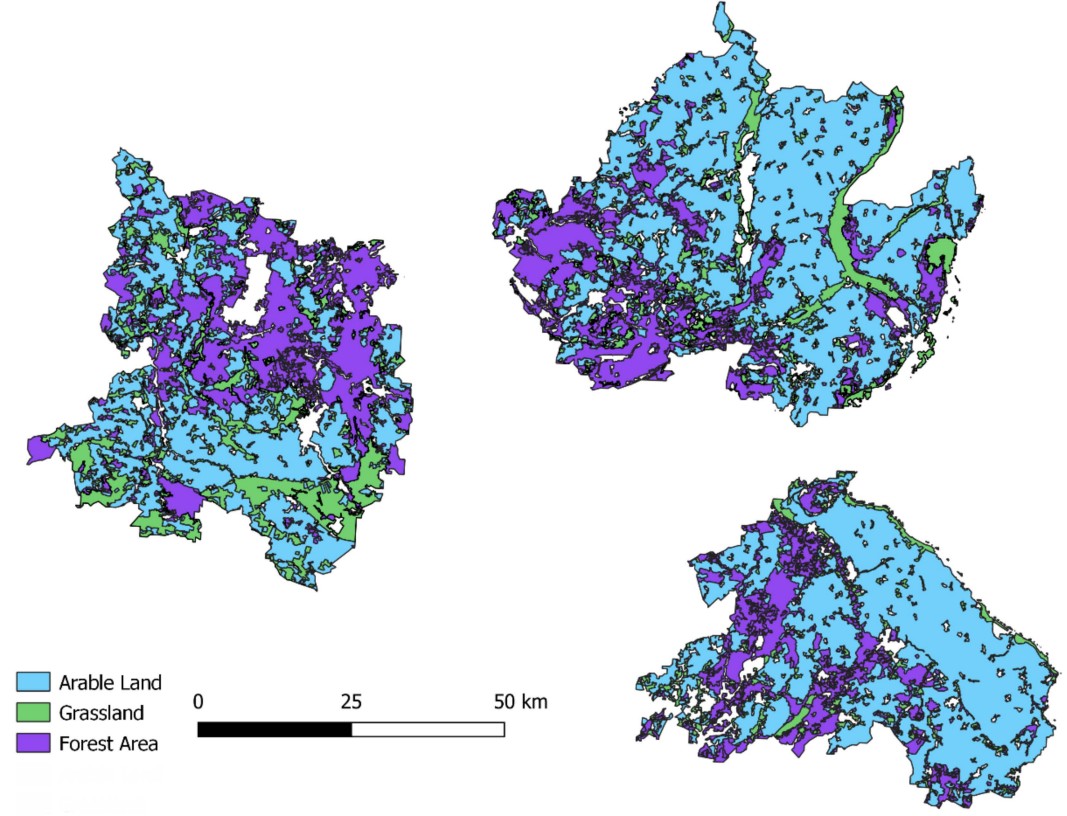

**Figure 1.** Land use in all case study areas.

Märkisch-Oderland (MOL) covers an area of 2158.67 km$^2$. The area used for agriculture in 2019 covered 1255 km$^2$ (58.14%) of the total area, of which 1160 km$^2$ were used for crops and 91 km$^2$ as permanent grasslands. Forest area comprises 510.27 km$^2$ or 23.6% of the total area [31].

Ostprignitz-Ruppin (OPR) covers 2509 km$^2$ and is located in the northwest of Brandenburg. It was founded in 1993, with the administrative center in Neuruppin. Agricultural and forest area cover 50 and 32%, respectively, of the total area. There are 18 nature reserves in OPR. Tourism, with possibilities for camping, hunting and water sports are important economic activities in addition to agriculture [32].

Uckermark (UM) covers an area of 3076.93 km$^2$ with 118,947 inhabitants [32]. In 2019, the agricultural area covered 1766 km$^2$ [27], of which 83% is used for crops, and 16.6% for grassland. Forest area covers 748 km$^2$ or 24% of the total area.

*2.2. Questionnaire Design*

We used Maptionnaire [33], a commercial PGIS tool that allows the active involvement of stakeholders in land use decision-making by capturing their perceptions. The tool has

the advantage of integrating survey questions with spatial data. It allows participants to indicate areas and spots of interest on an interactive map and relate these spatial data to specific questions and attributes. The questionnaire runs on a web platform with a specific URL for each survey. It was originally developed for efficient interaction with stakeholders and better informed decision-making in urban contexts, but has proven useful for research in rural areas [12] and landscape planning [9].

After an introduction to the goal of the survey and the topic of ES, participants were able to self-assign to different stakeholder categories. Consequently, the survey was structured by the five ES assessed. We chose five ES for the demand assessment, especially services related to soil functioning and yield. These are: (1) biodiversity, (2) carbon sequestration, (3) erosion control, (4) water availability and (5) yield. Regulatory services and functions are essential for maintaining the economic viability and long-term functioning of ecosystems [26]. We based our definition of the respective ES on the CICES [10]. The survey followed the same structure for each ES.

The ES was explained and agricultural management practices that influence the supply of the respective ES were mentioned in a detailed description. Participants were asked to self-assess their knowledge of these ES based on the previous explanation. Participants were asked in a mapping exercise to map one to three areas they consider relevant for the ES (Appendix A, Figure A1). Participants were asked to estimate the current perceived supply levels of the ES ("How do you estimate current supply levels?") and to then state their demand of the same ES as a percentage of the optimum state ("how high should the supply be?") within the same areas mapped by them in pop-up windows with closed-end-questions. This procedure was similar for all five ES and was followed by a brief section collecting demographic and socioeconomic data from the participants. At the end of the survey, participants had the possibility to evaluate the length, quality and relevance of the questionnaire.

The questionnaire consists of 28 pages. The core questions regarding the demand for ES take up only 10 pages, whereas the rest includes an introduction to the survey, consent to data protection, demographic classifications of the participants and evaluation of the questionnaire. Some questions were followed by sub-questions in pop-up windows, leading to a variation in the number of questions to be answered between 26 and 30, depending on the stakeholder group. Depending on the number of areas mapped, participants were able to answer between 19 (one area mapped per ES) and 47 questions (three areas mapped per ES) regarding the evaluation of ES. This leads to a total number of questions to be answered varying between a minimum of 26 and a maximum of 58.

There are four main advantages of this way of proceeding: firstly, the data collected contains information on the ecosystem the service is provided by, and on the beneficiary, i.e., the person by whom the demand is formulated. Secondly, it contains information about the gap between the current state perceived and the state demanded. The underlying assumption is that the formulated demand depends on the perception of the current state. Thirdly, it shows possible trade-offs that can arise if several non-synergetic ES are demanded in one area by displaying them in a spatial manner. Fourthly, it follows the concept by Geijzendorffer of distinction between the demand for goods marketed and goods desired.

### 2.3. Survey Dissemination and Scope of Sampling

The questionnaire was open from 1 March to 30 November 2020. The reason for this long period was the sudden outbreak of the COVID-19 pandemic and its impact on the availability of stakeholders and the impossibility of face-to-face visits.

After a pretest with selected participants, a thorough search for stakeholders in the region was conducted based on a spatial raster with previously defined categories related to our chosen ES—agriculture, forestry, nature conservation, tourism, inhabitants and others. The target audience were potential multipliers in our CSAs, i.e., people with a sufficiently large network and the possibility to distribute the questionnaire further,

from the different stakeholder groups in the areas identified. Management bodies of protected areas and environmental and agricultural associations were contacted in all three districts in Brandenburg, as well as organizations and action groups based in Berlin but working on a regional level. The questionnaire and the project were presented to them via email and telephone call. Furthermore, we distributed the questionnaire via social media channels and the homepage of the Leibniz Centre for Agricultural Landscape Research, and printed postcards with QR-codes directing those interested to the questionnaire. They were distributed in frequented places of the study regions during August 2020.

*2.4. Data Analysis*

The QGIS and R studio were used for the spatial and statistical analysis, respectively. Only completed questionnaires were included in the analysis. We conducted an overall analysis for the whole study region and analyzed subregions separately regarding interesting features and results.

2.4.1. Statistical Analysis

Using RStudio 1.3.959 [34] open source software, we estimate the correlation between stakeholder categories, supply perceived and demand stated for the five ES. The correlation between a nominal variable (stakeholder categories) and a continuous variable has been estimated through applying the function Intraclass Correlation Coefficient (ICCest) that uses the variance components from a one-way analysis of variance. The confidence interval is estimated by applying the type "THD", which is based upon the exact confidence limit equation in Searle [35] and can be used for unbalanced data (see Thomas and Hultquist [36]; Donner [37]). The aim of this analysis is to check whether actors from the same stakeholder category have a similar or close perception of the supply and demand or not. An example of the code used is shown in the text box below. This correlation has been further validated by coding the stakeholder categories with numerical values and implementing scatter plots between the respective coded stakeholder categories and supply perceived and demand stated.

```
Example of the code:
intraclasscc <- ICC::ICCest(Stakeholder_category,Water_Perceived_supply_average, data = NULL,
alpha = 0.05,CI.type = c("THD")) write.table(intraclasscc, 'cor.txt')
```

An average self-assessment of knowledge about the different ES was conducted for each CSA, and a ranking of the ES perceived as important common goods in the region. The relationship between the knowledge of ES and the perceived importance of ES was investigated.

The minimum, maximum and average values of the observations of the supply perceived $S_{avg}^k$ and demand stated $D_{avg}^k$ for each ES $k$ in each CSA as well as for the three CSAs together were calculated and consequently the respective absolute values of the gap $G_{avg}^k$ between supply perceived and demand stated was calculated according to Equation (1). We also created a boxplot based on the observations of the three CSAs regarding the same three variables to depict their dispersion across the median, 1st and 3rd quartile values. The boxplot has the advantage of showing outliers that also reflect poor knowledge and/or awareness that could otherwise be neglected consideration.

$$G_x^k = \left| S_x^k - D_x^k \right|, \text{ where } x = min, max \text{ or average} \tag{1}$$

2.4.2. Spatial Analysis

The areas mapped contain information about the extent of the area that participants estimate relevant for ES supply, and information on their state of ES supply perceived and their demand for the mapped areas. We analyzed the mapped areas in relation to the attributes assigned to the areas by the participants, and in relation to the land use they are mapped upon.

In the first step, the different areas mapped for each ES were overlaid and graduated colors were distributed according to the indicated level of ES perceived and demanded in

each area. For the same area, we calculated the gap between the supply and demand. We identified patterns of interest, and hotspots of demand and supply perceived in each CSA. We further focused on the visual identification of areas that were mapped for multiple ES, indicating potential trade-offs between ES supply.

In the second step, datasets on land use were integrated into the analysis. Data on arable land, grassland and forest from Corine Land Cover (CLC) 2018 [38] were used in order to assess the correlation between the areas mapped to different land uses. Specifically, the filtered categories were composed of (1) arable land: non-irrigated cropland (CLC code 211), (2) grassland: pasture and grassland (CLC code 231), and (3) forest: coniferous, deciduous and mixed forest (CLC codes 311, 312, 313, respectively). The areas mapped by the participants for each ES, were overlaid with the CLC 2018 maps. The overlaid areas were calculated in km$^2$ and % of the total mapped area.

### 2.5. Stakeholder Workshop

We discussed the results of the questionnaire with ten stakeholders from science, agriculture, entrepreneurship and regional management in an online stakeholder workshop. A focused open discussion emphasized the question regarding how participatory data can be used in regional management and landscape planning. We evaluated the criteria of successful participatory work and identified ways of implementing management strategies that can improve the supply of ES based on the assessment of demand.

## 3. Results

### 3.1. Respondents and Background

We collected a total of 53 complete questionnaires, of which 30 were collected in MOL, 14 in UM and 9 in OPR. The sample population was 40% male, 30% female, 2% diverse and 28% of the participants did not answer the gender question. Age distribution ranged between 21 and 80. Regarding affiliation with stakeholder groups, 34% selected farming and agriculture, 24% selected the stakeholder category of science, and 8% were related to civil society (Figure 2).

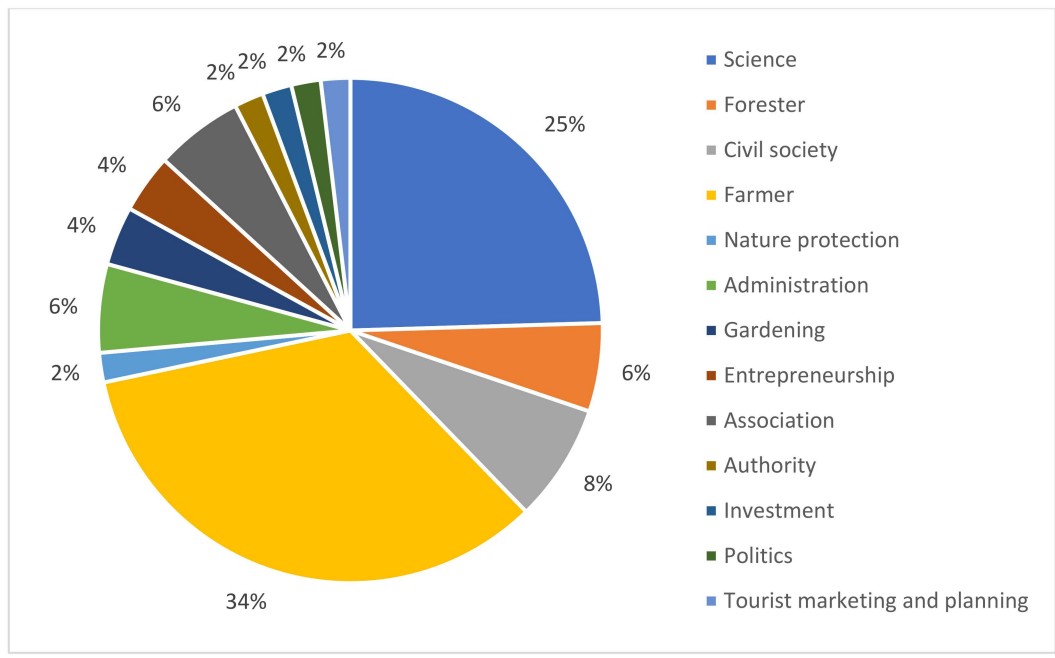

**Figure 2.** Percentage of stakeholder categories in the three case study areas.

Of the 18 farmers who completed the questionnaire, 10 reported working according to the guidelines of organic farming, 5 worked as conventional farmers, 1 was in transition to organic and 2 did not specify. Farm sizes varied between less than 20 ha (5 answers),

20–100 ha (7 answers), 100–500 ha (3 answers) and more than 500 ha (2 answers). The most frequent farm products are food crops, meat and vegetables (Appendix A, Figure A2).

### 3.2. Statistical Analysis

The intraclass correlation estimation showed a poor correlation between the variable stakeholder category against the average demand stated and the supply perceived of the different ES (Table 3). This be a consequence of the small sample size of actors from some stakeholder categories who took part in the survey. However, based on our results, no connection between the stakeholder group and the perception of current supply or demand for ES can be drawn. However, the highest intraclass correlation coefficient has been found with the supply of erosion control perceived, which has been checked via a scatter plot as shown in Figure 3 using numerical codes for stakeholder categories as shown in Table 4. It can be observed that farmers' supply of erosion control perceived falls to between 10 and 40%, whereas actors from science were split into two groups with high and low perception of the supply. Nevertheless, it ought to be noted that their mapped areas are not located in the same geographical location that justifies the variations in their perceptions, either.

**Table 3.** Intraclass correlation coefficients and confidence interval estimation results between stakeholder categories and supply perceived and demands for ES (CS = carbon sequestration; Bio = biodiversity; EC = erosion control).

| Stakeholder_Category | ICC | LowerCI | UpperCI | *n* | k | varw | vara |
|---|---|---|---|---|---|---|---|
| CS_Perceived_supply_average | −0.069 | −0.305 | 0.167 | 13 | 3.553 | 700.048 | −45.100 |
| CS_Demand_average | 0.024 | −0.173 | 0.378 | 13 | 3.553 | 1875.583 | 45.957 |
| Bio_Perceived_supply_average | 0.032 | −0.168 | 0.387 | 13 | 3.553 | 1001.460 | 32.876 |
| Bio_Demand_average | 0.013 | −0.180 | 0.365 | 13 | 3.553 | 2085.036 | 28.245 |
| Yield_Perceived_supply_average | −0.112 | −0.254 | 0.196 | 13 | 3.553 | 1228.897 | −123.892 |
| Yield_Demand_average | −0.009 | −0.194 | 0.339 | 13 | 3.553 | 1844.923 | −15.690 |
| Water_Perceived_supply_average | −0.075 | −0.233 | 0.250 | 13 | 3.553 | 652.103 | −45.432 |
| Water_Demand_average | −0.073 | −0.232 | 0.253 | 13 | 3.553 | 1864.221 | −126.847 |
| EC_Perceived_supply_average | 0.157 | −0.084 | 0.518 | 13 | 3.553 | 555.733 | 103.313 |
| EC_Demand_average | −0.001 | −0.189 | 0.348 | 13 | 3.553 | 1573.915 | −0.861 |

Abbreviation–Definition: ICC—intraclass correlation coefficient; LowerCI—lower confidence interval limit, where the confidence level is set by alpha; UpperCI—upper confidence interval limit, where the confidence level is set by alpha; *n*—total number of individuals or groups used in the analysis; k—number of measurements per individual or group. In an unbalanced design, k is always less than the mean number of measurements per individual/group and is calculated using the equation in Lessells and Boag [39]; varw—within individual or group variance; vara—among individual or group variance.

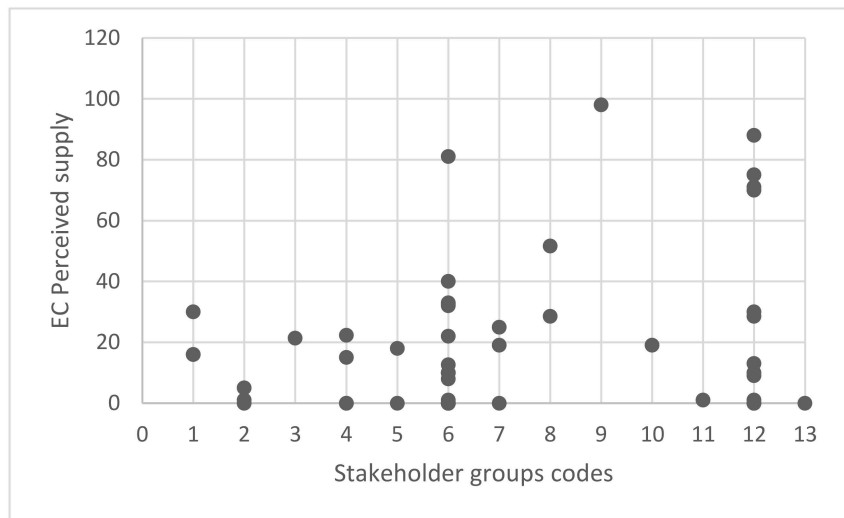

**Figure 3.** Scatter plot between the supply of erosion control (EC) perceived and coded stakeholder groups to display the correlation between the two variables.

**Table 4.** Codes of stakeholder groups used for the scatter chart in Figure 3.

| Stakeholder Category | Code | Stakeholder Category | Code |
|---|---|---|---|
| Administration | 1 | Gardening | 8 |
| Association | 2 | Investment | 9 |
| Authority | 3 | Nature protection | 10 |
| Civil society | 4 | Politics | 11 |
| Entrepreneurship | 5 | Science | 12 |
| Farmer | 6 | Tourist marketing and planning | 13 |
| Forester | 7 | N.B. Zero value = not answered | |

Average values of knowledge self-assessment for the different ES ranked between 60 and 70%, where biodiversity and water availability scored highest with 70%, and carbon sequestration lowest with 62.5%. Average awareness in UM and OPR was highest for erosion control with 79.9 and 76.8%, respectively, whereas water availability scored highest in MOL with 73%. Biodiversity and water availability were chosen most often in the ranking of ES as important public goods with 31 and 34 votes, respectively. Again, erosion control scored higher in OPR and UM than in MOL.

The average demand stated exceeded the average supply perceived for all ES in the test regions. Current supply levels were estimated to be the lowest on average for erosion control (24% of the optimum state), while biodiversity and yield were estimated to have higher supply levels with 47 and 55% of the optimum, respectively. Water availability scored relatively low in MOL (31%) and UM (32%), and relatively high in OPR (48%), leading to an average value of 33%, the same as carbon sequestration (33%). Average demand values stated were lowest for yield (79%), carbon sequestration (79%) and erosion control (80%), and highest for biodiversity (85%) and water availability (89%). The average supply demand gap was highest for erosion control and water availability (56%) and lowest for yield (24%) (Appendix A, Figures A3 and A4). The range of supply perceived is rather wide, from 24 to 55 average percentage points, whereas demands stated are at least 24% higher than supplies, and much narrower, ranging from 79 to 89%.

Figure 4 shows a boxplot comparing the supply perceived, demand stated and the gap for all ES values. The boxplot shows the dispersion of data for the three variables, which are stretched between 0 and 100 for almost all ES. However, the median value of the supply perceived is about 20% for carbon sequestration, water availability and erosion control, which is also very close to the 1st quartile value, indicating an agreement on a narrow value between 10 and 20%. Meanwhile, the demands for these ES comes in between 70 and 85%, which is justified by a gap ranging from 40 to 60%. On the contrary, it is more likely to lie between 60 and 75% for the supply of yield perceived with a close value of demand (75–80%), which makes the gap very small (about 20%). Surprisingly, the demand for biodiversity shows a high agreement between 90 and 100% and a low perceived supply (20–40%). The high dispersion between the median and the 1st or 3rd quartile or between the minimum or maximum value and the 1st or 3rd quartile, respectively, suggests some outlier data that might have originated from the low awareness of the participants about the status of the ES in the CSAs.

### 3.3. Spatial Analysis

#### 3.3.1. Demand Area Index

The analysis of the areas indicated by respondents and ES was carried out for each district separately and aggregated over participants. We obtained information on the supply perceived and demand stated for the same area, and calculated the gap between supply and demand. Some areas or parts of an area were mapped several times by different participants or for different ES and were counted as often as they were mapped in this analysis. Only areas within the questionnaire boundaries were used for the spatial

evaluation. The demand area index captures the total added area of all areas mapped. Table 5 shows these overall area values as an index for the three CSAs in km$^2$.

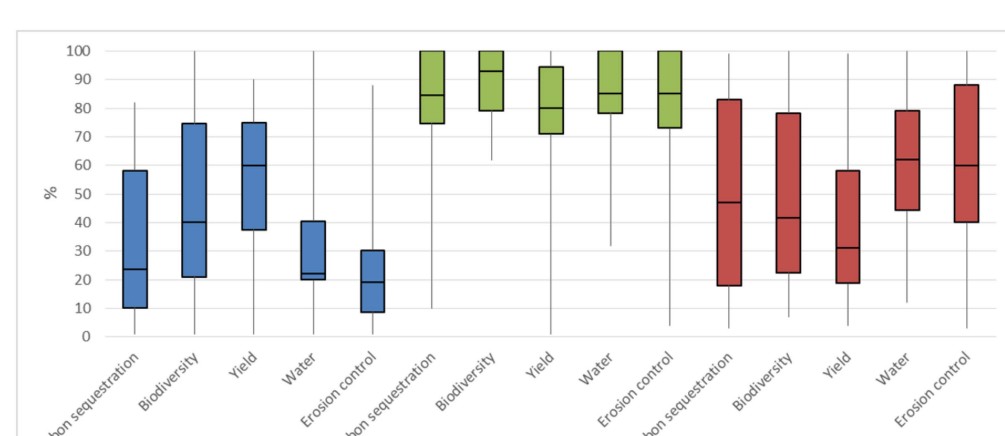

**Figure 4.** Average supply and demand assessment for ES in all three CSAs.

**Table 5.** Area mapped in km$^2$.

|  | MOL (km$^2$) | % of Total Area Mapped | OPR (km$^2$) | % of Total Area Mapped | UM (km$^2$) | % of Total Area Mapped |
|---|---|---|---|---|---|---|
| Biodiversity | 21,551.5 | 13.67 | 10,212.7 | 16.19 | 8266.94 | 24.88 |
| Carbon Sequestration | 9710.42 | 6.16 | 25,911.9 | 41.08 | 2333.75 | 7.02 |
| Erosion Control | 41,159.3 | 26.10 | 6358.08 | 10.08 | 12234 | 36.82 |
| Water Availability | 69,534.8 | 44.09 | 15,032.2 | 23.83 | 7482.88 | 22.52 |
| Yield | 15,739 | 9.98 | 5559.28 | 8.81 | 2911.41 | 8.76 |
| Total area mapped (km$^2$) | 157,695.02 |  | 63,074.16 |  | 33,228.98 |  |

The total area mapped was largest in MOL due to the high number of participants. The ES with the largest share mapped was water availability in MOL, with 44% of the total area mapped, followed by erosion control (26%) and biodiversity (14%). The largest area in OPR was mapped for carbon sequestration (41%), followed by water availability (24%) and biodiversity (16%). Erosion control (36%), biodiversity (25%) and water availability (22%) were the areas with the largest overall surface in UM. Yield scored below 10% of the total area mapped in all three districts.

Example Erosion Control in MOL

We use the results from the CSA MOL and the ES erosion control exemplarily to display relationships between the demand stated, supply perceived and the mapped areas.

The level of supply perceived indicated in MOL varied between 1 and 88 on a scale from 0 to 100% (Figure 5a). A total of 32 out of 45 participants (71%) rated current supply levels at 50 or lower, 4 participants (8%) rated supply levels higher than 50, and 9 participants (20%) only mapped an area without answering the question on supply levels.

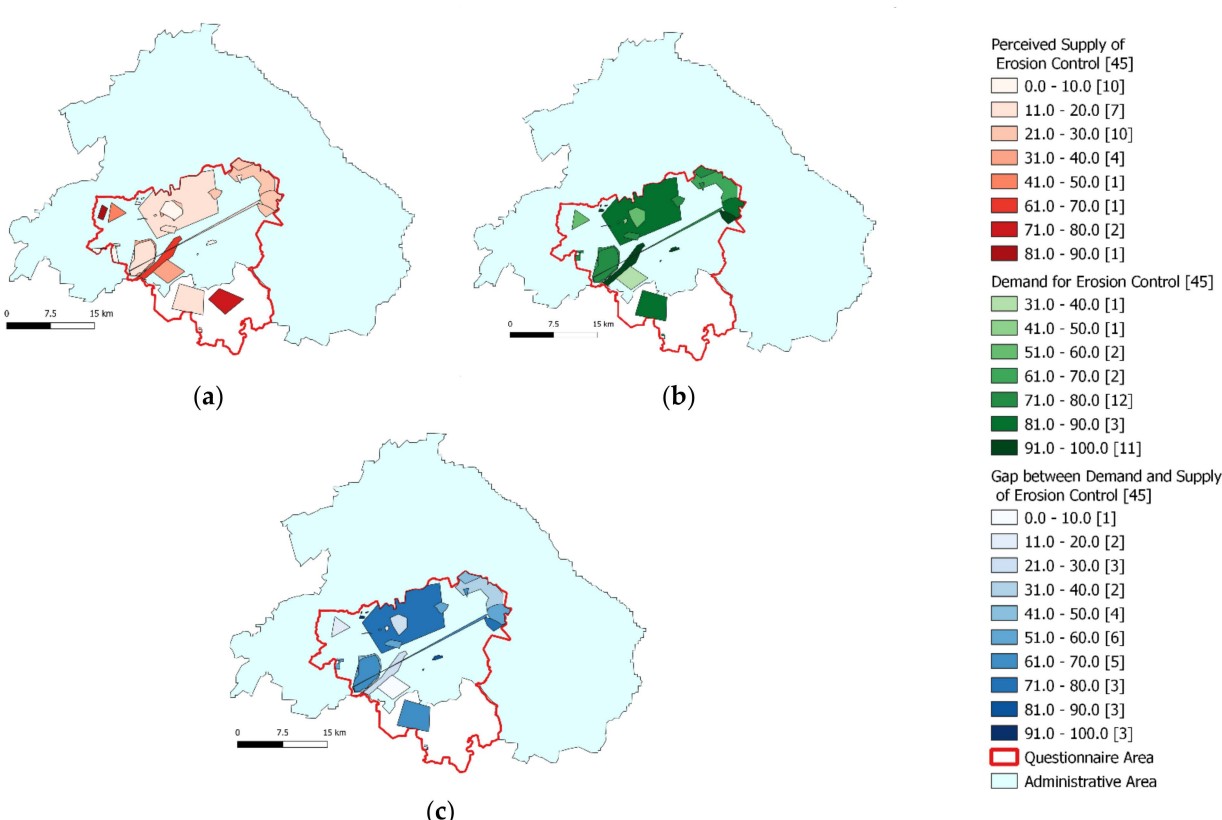

**Figure 5.** Supply perceived (**a**), demand stated (**b**) and the gap (**c**) for erosion control in MOL.

The levels of demand stated ranged from 40 to 100 (Figure 5b), where 30 out of 45 respondents (67%) rated their demand higher than 50, participants (4%) found their demand to be below 50, and 13 participants (29%) did not give an indication about their demand.

The demand levels indicated were in all cases higher than the supply levels perceived. The gap between the supply perceived and demand stated ranged from 7 to 98 (Figure 5c). The gap between the current state and the state demanded was higher than 50 in 20 of 45 cases (44%). Respondents found the discrepancy between demand and supply to be lower than 50 in 12 cases (27%). Thirteen respondents did not indicate either their demand or supply perceived, so the gap could not be calculated. Figure 5a–c show that the gap between demand and supply is highest where the supply perceived is rated rather low (<30%), and the demand rather high (>60%). Small gaps between demand and supply occur where both the state perceived and the demand are high or low. However, in a few areas the current state perceived equals the demand stated. The results do not show hot spots of demand for erosion control. However, in combination with demand maps for other ES, overlays of demand for multiple ES in almost all regions can be identified.

### 3.3.2. Mapped Area and Land Use

We again use the results from the CSA MOL, especially for erosion control, to demonstrate results for the analysis of the relationship between mapped areas and land use. Arable land covers 1160 km$^2$ or 53.7% of the total area in MOL. Results show that the largest areas mapped lies on arable land (Figure 6). This reflects the larger share of arable land in the landscape and is also an indication that the awareness of ES is higher in arable land than in forest and grassland areas. Table 6 shows the surface of the areas mapped by the participants in MOL for each ES, and its distribution by land use type. Areas mapped were only calculated once and values reflect the surface area mapped. Participants mapped proportionally more area on arable land than on forest or grassland sites. The

share mapped on arable land for all ES was between 41 and 66% of the total area mapped. An average of between 5 and 14% were mapped on grassland, and between 18 and 39% on forest area. This reflects the general distribution of arable land, which is comparably higher than grassland and forest in all three CSAs (see Section 2.1), and even exceeds the proportion of arable land on the total area. The share mapped on arable land for some ES clearly exceeds the general share of arable land due to a coupled assignment of spatial entities to different ES, which can also be interpreted as an indication of the increased awareness of potential trade-offs between yield and other ES on arable land.

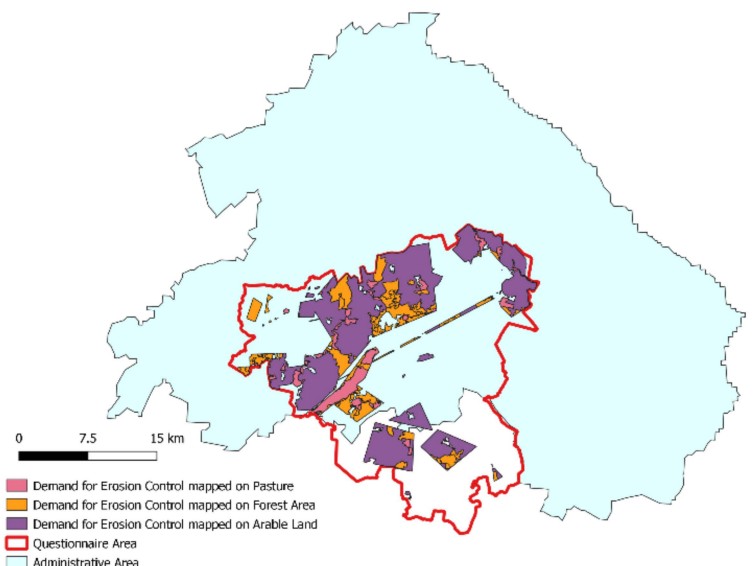

**Figure 6.** Demand for Erosion Control and Land Use in MOL.

**Table 6.** Mapped area for all ES per land use in Märkisch-Oderland.

| | Biodiversity (km²) | % of Total Surface Area Mapped | Carbon Seques-tration (km²) | % of Total Surface Area Mapped | Erosion Control (km²) | % of Total Surface Area Mapped | Water Avail-ability (km²) | % of Total Surface Area Mapped | Yield (km²) | % of Total Surface Area Mapped |
|---|---|---|---|---|---|---|---|---|---|---|
| Arable land | 7294.76 | 40.80 | 5505.98 | 64.12 | 15,738.6 | 63.30 | 23,261.8 | 49.09 | 6970.03 | 65.60 |
| Pasture | 1836.81 | 10.27 | 1172.54 | 13.66 | 2223.18 | 8.94 | 3598.08 | 7.59 | 507.87 | 4.78 |
| Forest area | 6868.79 | 38.41 | 1526.91 | 17.78 | 5083.63 | 20.45 | 15,358.5 | 32.41 | 2794.77 | 26.30 |
| Total surface area (km²) | 17,881 | | 8586.82 | | 24,864.8 | | 47,388.1 | | 10,625.2 | |

The largest areas in MOL are mapped for water availability, followed by erosion control and biodiversity.

### 3.3.3. Stakeholder Workshop

Our intention in conducting an online workshop with ten local representatives of the stakeholder groups addressed in the survey was to validate how far the results from this demand mapping exercise could be useful in regional and landscape management. Four general conclusions could be drawn from the stakeholder side. (1) A systemic approach combining the generation of knowledge on ES and the communication and dissemination of this knowledge is necessary. Participatory tools can be a helpful vehicle for improving the collection of perceptions, but the overarching goal has to be a more systemic perspective of management for a multitude of ES. (2) Trade-offs impede the achievement of measures to improve multiple ES. The PGIS can help to identify parts of these trade-offs. (3) The possibilities and limitations of participation should be clarified when inviting stakeholders to take part. Motivation decreases if people are asked for their opinion but their suggestions find no reflections in reality. (4) The more explicit the goal of

participation, the more targeted a manner in which it can be carried out. The use of PGIS in workshops with technical guidance could be useful in regional development strategies or for landscape planning and decision-making in municipalities.

## 4. Discussion

In this discussion, we critically review our study results. In order to find out how this research could be applied in other areas, we then work out interfaces with other studies.

The results suggest that there is no difference between the supplies demanded and perceived according to the stakeholder group. Local people show similar understandings of the surrounding landscape, independent of their profession or affiliation. Furthermore, demands exceed the supply perceived of all ES in all CSAs. For all ES, participants suggest that the current levels do not match the state needed for an ecological equilibrium. The gap between demand stated and supply perceived is highest for erosion control and water availability (56%) and lowest for yield (24%). Participants mapped less than 10% of the total area mapped for yield in all three CSAs. Biodiversity and water availability are the two ES most demanded. Both are also the ES most recognized by the participants as important public goods. In addition, participants report having a generally high knowledge about these ES. Higher self-assessed knowledge can lead to a higher awareness of demand for the respective ES.

The comparison of areas mapped and current land use on the mapped areas indicates an increased awareness of ES on arable land in comparison to grassland and forest area. The share of areas mapped on arable land exceeds the share of arable land in the CSAs for most ES. This result is in line with the problem focus on erosion control and water availability, both features of arable land that particularly emerge with rather uniform cropping patterns and low diversification of landscape and crop rotations. This causal explanation has also been raised by the stakeholder side when presenting and discussing these results.

The focus of our empirical study is to attempt to locate and evaluate other categories of ES both spatially and explicitly using PGIS. Since a comparable direct subjective level of experience or perception, as with cultural dimensions (e.g., beauty, harmony, naturalness), cannot be assumed here, we allowed all respondents equal information access to the queried ES and functions—biodiversity, carbon sequestration, erosion control, water availability, yield—and to infer indications of their condition.

When interpreting the results, it is important to keep in mind that experience ratings and information may have interacted to varying degrees here. It cannot be excluded, for example, that the ES erosion control and water availability are mentioned primarily because their assessment can be more observation-based than, for instance, the more abstract knowledge-based ES carbon sequestration.

At the same time, at the very least, we felt it was important and necessary to contrast the distribution of mapped ES demand preferences and ES supply situations perceived for the purposes of consistency analysis according to Brown et al. [40]. These authors mention spatial accuracy and credibility (reputation, trustworthiness and motivations of the spatial data contributor) as key data quality features for the use of public participatory mapping in land use planning. We addressed spatial accuracy with the proof of logical consistency with land use classes distribution ('validity-as-accuracy'). Regarding 'validity-as-credibility,' we did not find any difference in the outcome depending on who did the assessment.

The use of participatory instruments can be a powerful tool for democratizing land use planning and bringing transparency into decisions. The PGIS can be a useful tool for investigating people's perception of ES in the landscape and capturing their desired state. However, possible pitfalls should be considered and avoided. In particular, the reason for participation should be clarified, and the possibilities and limitations should be made clear to the participants. The format of map-based identification of very specific landscape requires much knowledge of the landscape itself and elaborate digital skills. A more targeted way in larger rural areas might be to use this tool in workshops or in smaller

groups of local experts under technical guidance. This would allow the collection of more background information on location-specific perceptions and notions of individuals, as well as discursive statements between experts. Such could provide a valuable tool for local land use governance processes.

PGIS and stakeholder analysis could in further research be combined with geostatistical or modeling approaches for obtaining robust and site-specific results. Simulation models or optimization models have the advantage of allowing quantification, upscaling and systemic assessments. However, leaving out stakeholder involvements in these assessments leads to limited practical utility for site-specific decision making [41]. The combination of both participatory mapping and analytical or modeling approaches can produce integrated and practical results for land-use decisions, both in trade-off analyses and in ES valuation.

The combination of participatory mapping of non-cultural ES with monetary valuations is another field of potential use. The reason why the empirical focus of social needs has been predominantly on cultural ES so far is that the preferences stated can also be expressed most clearly here, since they are based on individual evaluations of experience. Fagerholm and Käyhkö [42] describe social landscape evaluations as subjectively experienced and related to location as well as context. Monetary valuations of ES can give an idea about their value in relation to societies' gross domestic product but often miss out on functionalities and system dynamics that are not yet understood. Furthermore, they foster an idea of the replaceability of ES by financial means. The combination of mapping with monetary approaches could capture both numerical and spatial values that enable a closer approximation of the economic and social value of the ES. A study by Kenter [43] combines choice experiment with participatory systems modelling, participatory mapping and psychometric analysis. Results show that with a participatory component, participants were better able to include personal values, a public perspective or place-based values into the monetary valuation of ES and the analysis of trade-offs [43].

The use of decision support systems (DSS) play an increasing role in the choice of management strategies for the supply of multiple ES, climate change mitigation and biodiversity preservation in agriculture and forestry [44,45]. The challenge of balancing different demands for ES in agricultural landscapes creates a necessity for DSS that integrate demands for provisioning, regulating and cultural ES across spatial and temporal dimensions. The usefulness of results from ES demand and supply assessment for DSS requires understandable data representation. More research is needed on the question of how ES information can be integrated into DSS in a decision-supportive way [46]. Our research shows one possibility of representing demand in a spatially explicit way that could be integrated with biophysical data on supply and recommendations for management in DSS.

## 5. Conclusions

We used a map-based questionnaire to collect data on demand and supply perception of five ES, formulated by different stakeholder groups in three CSAs. We discussed the results of the survey with experts from the stakeholder groups in a stakeholder workshop. Our aim is to evaluate the usefulness of PGIS methods in larger rural and agrarian contexts and to contribute a new methodological approach for assessing spatially explicit demands for regulating and provisioning ES. The demands for ES play a growing role in the management of ES, especially in agricultural areas. A harmonization of different demands from different stakeholders can avoid trade-offs and alleviate the decision for management strategies to improve multiple ES supply.

Our study shows no significant differences in demands between stakeholder groups. Our results rather suggest the importance of including local knowledge on landscapes in land use decisions and give the first indication that people from different stakeholder categories have profound knowledge of their surrounding ecosystems.

These results are preliminary, and we encourage further systematic investigation into its procedural aspects. We recommend the methodology presented as a starting point

for demand analyses in similar agrarian contexts to generate results for comparison. The issue of spatial trade-off mapping is a valuable investigation area, where we recommend a smaller scale and reduced number of ES with a rather distinct functional interrelation as a starting point. This research could be extended by complementary geostatistical or analytical approaches.

**Author Contributions:** Conceptualization, A.P., C.S. and M.S.; methodology: Section 2.1, C.S.; Section 2.2, C.S. and A.P.; Section 2.3, C.S. and A.P., Section 2.4.1, M.S.; Section 2.4.2, C.S. resources, A.P. and S.D.B.-K.; writing—original draft preparation, C.S.; writing—review and editing, A.P., M.S. and S.D.B.-K.; visualization, C.S. and M.S.; supervision, A.P and S.D.B.-K.; project administration, A.P. and S.D.B.-K.; funding acquisition, A.P. and S.D.B.-K. All authors have read and agreed to the published version of the manuscript.

**Funding:** This research was funded by the German Federal Ministry of Education and Research (BMBF) through the Digital Agriculture Knowledge and Information System (DAKIS) Project, grant number FKZ 031B0729A, and the APC was funded by the DAKIS Project.

**Institutional Review Board Statement:** Not applicable.

**Informed Consent Statement:** Not applicable.

**Acknowledgments:** We would like to thank all reviewers for their efforts and fruitful feedback on the article.

**Conflicts of Interest:** The authors declare no conflict of interest. The funders had no role in the design of the study; in the collection, analyses, or interpretation of data; in the writing of the manuscript; or in the decision to publish the results.

## Appendix A

*Appendix A.1. Ecosystem Services*

In our study, we focus on the mapping of five regulating ES in the landscapes of Brandenburg. The selection of ES follows the typology of CICES and includes four regulating and one provisioning ES. The descriptions we gave in the questionnaire of the five ES chosen were as follows.

*Appendix A.2. Erosion Risk and Erosion Control*

Soil erosion describes the removal of fine-grained topsoil by wind or water leading to a deterioration in the soil quality. Erosion is a natural process but is often greatly enhanced or triggered by the use of soils. Negative impacts of erosion can range from a reduction in soil fertility, loss of important soil functions and crop failure to flooding and contamination of trails and roads. The erosion risk of sites is largely determined by natural factors, such as slope, rainfall intensity and soil characteristics. Land use by humans, particularly the geometry and size of cropland, selection of crop types and intensity of tillage, can influence erosion risk significantly. Appropriate erosion control measures include establishing diverse crop rotations, ensuring long soil cover through cover crops and undersowing, slope-parallel tillage, and conservation tillage, for example, mulch seeding. At the landscape level, windbreak plantings in the form of hedges and woody elements and the permanent greening of slope hollows and depth contours up to change of use in areas particularly at risk of erosion can be useful.

*Appendix A.3. Water Availability*

Almost 98% of agricultural land in Germany is fed by green water, i.e., rainwater. The amount of groundwater and surface water available to plants is influenced by meteorological and hydrological factors. Climate change is altering the dynamics of the temporal availability of rainwater so that periods of heavy rainfall and drought may occur. Rising temperatures also favor evaporation rates. At the same time, topography and soil characteristics affect a soil's ability to hold plant-available water. Appropriate land use measures

can influence the average water availability positively—the soil's water-holding capacity can be improved by building up soil organic carbon. Crop and variety selection can be adapted using varieties with lower water requirements.

### Appendix A.4. Carbon Sequestration

Soil organic matter is around 50% carbon and is an important feature of soil fertility. Soils with a high organic matter content can store and release more nutrients and water to plants than soils with less organic matter. In addition, carbon sequestration in soils is increasingly seen as a way of reducing atmospheric carbon levels as mitigation and adaptation strategies for global climate change. Increasing (anthropogenic) carbon sequestration in soils and plants can be achieved by:

- favoring biomass growth. Perennial crops and woody plants, in particular the removal of the greenhouse gas $CO_2$ from the atmosphere by building up biomass, thus contributing to climate change mitigation.
- the development of organic matter-rich soil horizons by adding organic material (compost, crop residues).

### Appendix A.5. Biodiversity

Biological diversity or biodiversity ensures the vital services provided by nature. Biodiversity encompasses all living things in their various habitats: in the soil, in water and on land—from animals and plants to fungi and bacteria. In addition to species diversity, biodiversity includes genetic diversity and the diversity of communities of organisms. Biodiversity provides key regulatory services to ecosystems, such as the pollination of crops, soil fertility, protection against environmental disasters, such as floods, landslides and avalanches, purification of water and air, decomposition of waste and pollutants, and natural pest control. Human interference with nature alters the food base and habitats of organisms and thus ecosystems. Urban sprawl, landscape fragmentation and increasing land use are negatively affecting the number of habitats. At the same time, land use holds great potential for implementing biodiversity-enhancing measures. These include the creation of large reservoirs of biodiversity of farm animals, cultivated plants, habitats and wild organisms adapted to them.

### Appendix A.6. Yield

Yield is the biomass of crops and fodder plants harvested and marketed per year per hectare. On arable land, this refers to food crops, energy crops, fodder crops and industrial crops. On forestry land, yield refers to the biomass of wood.

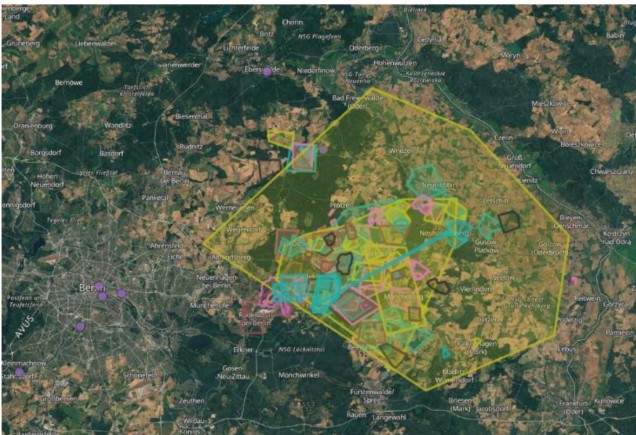

**Figure A1.** Screenshot of Maptionnaire survey surface.

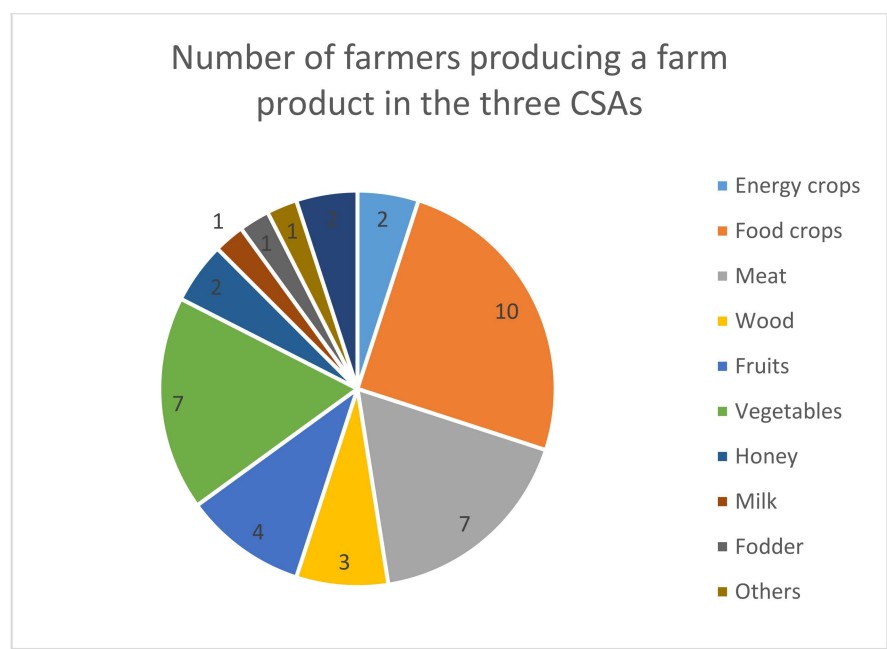

**Figure A2.** Variety of farm products produced in all three CSAs.

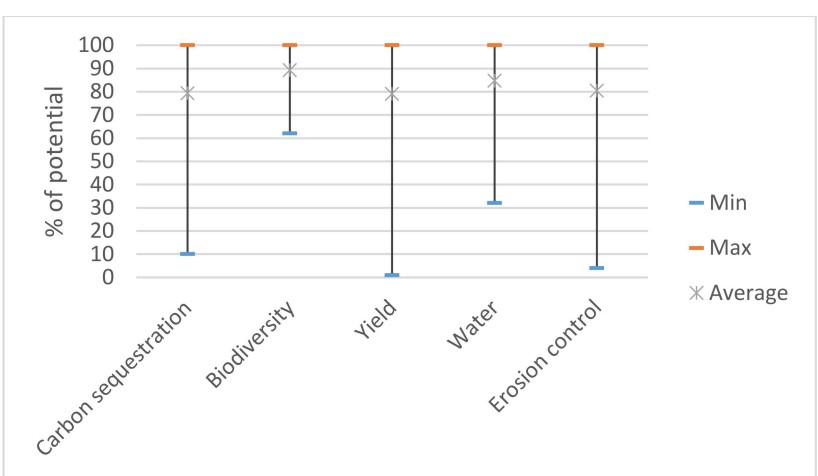

**Figure A3.** Demand stated for ES in all three CSAs.

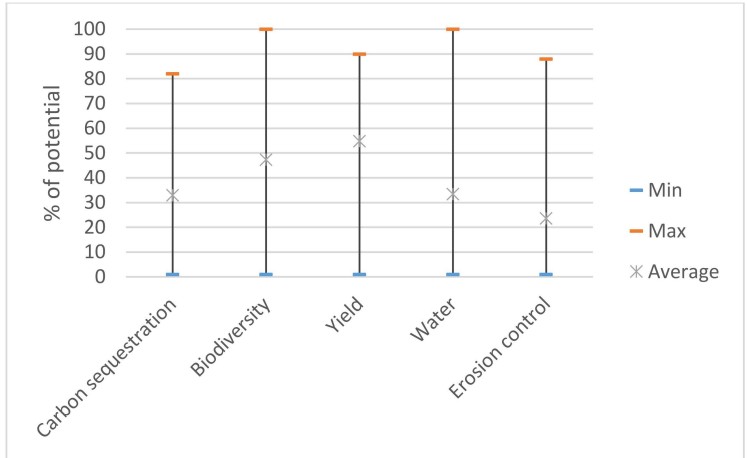

**Figure A4.** Supply perceived for ES in all three CSAs.

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
