# Peer review of "Participatory Mapping of Demand for Ecosystem Services in Agricultural Landscapes"

_agriculture, doi:10.3390/agriculture11121193_

Round 1

Reviewer 1 Report

This manuscript has an interesting topic about how PPGIS could help to assess the demand of ecosystem services in larger rural and agrarian contexts. The specific questions was as follows:                                                 (1) Introduction: the concept of 'trade-offs' that usually in use of trade-off and synergy in multiple ecosystem services supply. Please explain deeper why it is also vital in this ES supply-demand relations research. 

(2) Methodology: the selection of five ecosystem services should be further explained that why they are vitial for the this kind of research. Specifically, 'Biodiversity' is a 'function' but not a 'service' in the CICES scheme which has distinguished the 'function', 'service' and 'benefits'.

(3) Discussion: Please explain the significance that how this research could be applied in other areas by comparing with other studies.

Author Response

Thank you for your comments and your thorough read. Please find attached answers to your comments. In the manuscript, please have a look at the respective position to see the corrections.

  1. We added a section in the introduction with the title 1.3 Assessment of regulating services, and added several phrases in the section 1.2 demand assessments and trade-offs, to point out the importance of trade-off analysis in supply-demand relations research.
  2. We clarified the difference between service and function and added a more detailed description of the CICES framework in the introduction. Now the paragraph on CICES in the method section refers to the explanation given in the introduction.
  3. Thank you for pointing this out. Please find in the conclusion part now several changes. We separated the “Discussion” and “Conclusion” chapter, and included in the conclusion three small paragraphs on potential use of the study results in further studies. Our methodology could be adapted and further developed for the choice of management strategies with decision support systems. We further highlight the importance of developing integrative approaches of combining participatory with analytical and modelling approaches, and point to the importance of participatory tools also in monetary valuation especially of regulating ES.

Reviewer 2 Report

This is an interesting paper, although rather preliminary in demonstrating its potential. The authors address the demand side of ecosystem services through Participatory Geographic Information System methods. This resulted in a statistically-based, geospatial assessment of the needs across five different agricultural landscapes in Northeast Germany and provides a potential approach to developing data for decision support systems. As noted by the authors, the reported research is highly preliminary but this topic is of great interest to many involved with land-use, and it may help flag where others can make improvements on this topic. Nearly all of my comments/edits were associated with the Introduction section. It is almost as if a different person wrote it, compared to the rest of the manuscript. See attached file for further information.

Author Response

Thank you for your comments and your thorough read. Please find attached as a PDF answers to your comments.

Round 2

Reviewer 1 Report

What the authors did  in the revised version did increase the quality of the paper. One suggestion in discussion was recommended: adding references for comparable results in related studies will be better.

Author Response

Dear Sir or Madam,

Thank you for your comment. Please find the revised manuscript, with references for comparable results in the discussion, and a shortened conclusion.